# Capsaicin Inhibits Multiple Voltage-Gated Ion Channels in Rabbit Ventricular Cardiomyocytes in TRPV1-Independent Manner

**DOI:** 10.3390/ph15101187

**Published:** 2022-09-26

**Authors:** Dmytro Isaev, Keun-Hang Susan Yang, Waheed Shabbir, Frank Christopher Howarth, Murat Oz

**Affiliations:** 1Department of Cellular Membranology, Bogomoletz Institute of Physiology, 01024 Kiev, Ukraine; 2Department of Biological Sciences, Schmid College of Science and Technology, Chapman University, One University Drive, Orange, CA 92866, USA; 3Department of Physiology, College of Medicine and Health Sciences, UAE University, Abu Dhabi 15551, United Arab Emirates; 4Department of Pharmacology and Therapeutics, Faculty of Pharmacy, Kuwait University, Safat 13110, Kuwait

**Keywords:** capsaicin, ion channels, rabbit, cardiomyocytes

## Abstract

Capsaicin is a naturally occurring alkaloid derived from chili pepper which is responsible for its hot, pungent taste. It exerts multiple pharmacological actions, including pain-relieving, anti-cancer, anti-inflammatory, anti-obesity, and antioxidant effects. Previous studies have shown that capsaicin significantly affects the contractility and automaticity of the heart and alters cardiovascular functions. In this study, the effects of capsaicin were investigated on voltage-gated ion currents in rabbit ventricular myocytes. Capsaicin inhibited rapidly activated (*I*_Kr_) and slowly activated (*I*_Ks_) K^+^ currents and transient outward (*I*_to_) K^+^ current with IC_50_ values of 3.4 µM,14.7 µM, and 9.6 µM, respectively. In addition, capsaicin, at higher concentrations, suppressed voltage-gated Na^+^ and Ca^2+^ currents and inward rectifier *I*_K1_ current with IC_50_ values of 42.7 µM, 34.9 µM, and 38.8 µM, respectively. Capsaicin inhibitions of *I*_Na_, *I*_L-Ca_, *I*_Kr_, *I*_Ks_, *I*_to_, and *I*_K1_ were not reversed in the presence of capsazepine (3 µM), a TRPV1 antagonist. The inhibitory effects of capsaicin on these currents developed gradually, reaching steady-state levels within 3 to 6 min, and the recoveries were usually incomplete during washout. In concentration-inhibition curves, apparent Hill coefficients higher than unity suggested multiple interaction sites of capsaicin on these channels. Collectively, these findings indicate that capsaicin affects cardiac electrophysiology by acting on a diverse range of ion channels and suggest that caution should be exercised when capsaicin is administered to carriers of cardiac channelopathies or to individuals with arrhythmia-prone conditions, such as ischemic heart diseases.

## 1. Introduction

Capsaicin (8-methyl-N-vanillyl-6-nonenamide), as an algogen and prototypical activator of transient receptor potential vanilloid type 1 (TRPV1) channels, has been used extensively in pain research. In addition, capsaicin has been shown to play functional roles in a wide range of pathophysiological conditions, ranging from cardiovascular to respiratory and urinary diseases [1,2]. The effects of capsaicin on the cardiovascular system have been known for several decades. Activation of TRPV1 channels on capsaicin-sensitive sensory nerves in the cardiovascular system has been shown to release various neuropeptides and play important roles in physiological regulation, as well as the pathophysiology of cardiovascular diseases such as heart failure, myocardial infarction, and hypertension [3,4]. The vascular effects of capsaicin and the role of TRPV1 channels in the pathophysiology of hypertension is a complex process involving the TRPV1-dependent and independent action of capsaicin on the vascular endothelium, smooth muscle myocytes, and the peripheral and central nervous system, which have been reviewed in earlier studies [3,5,6,7].

In earlier studies, capsaicin has been shown to exert transient positive inotropic and chronotropic effects [8,9,10], which are reversed by TRPV1 antagonists such as ruthenium-red [11,12] and capsazepine [12] and by the desensitization of TRPV1 [13,14] in isolated guinea-pig and rat atria. In vitro studies have demonstrated that positive inotropic and chronotropic effects of capsaicin (0.1–1 µM) are mediated by neuropeptides such as calcitonin gene-related peptide, neurokinin A, and vasoactive intestinal polypeptide released from capsaicin-sensitive-sensory neurons in guinea-pig- and rat-isolated atrial preparations [11,12,14,15,16].

Although previous studies have established a functional role of TRPV1 channels in mediating the positive inotropic and chronotropic effects of capsaicin, mainly in atrial preparations, the capsaicin-induced suppression of cardiac contractions and alterations in electrophysiological characteristics of action potentials have also been reported in ventricular papillary muscle and [10,17], whole-heart preparations [11,16,18,19], suggesting that capsaicin, in addition to TRPV1-mediated effects, exerts direct actions on the cardiac muscle. Thus, this study investigated the effects of capsaicin on major inward (Na^+^ and Ca^2+^) and outward (K^+^) currents in rabbit ventricular myocytes, which are physiologically the most relevant cells to investigate the cellular and molecular targets of capsaicin actions observed in earlier studies in atrial and ventricular preparations.

## 2. Results

### 2.1. The Effects of Capsaicin on Na^+^ Channels (I_Na_)

Inward *I*_Na_ was elicited by step depolarizations from a holding potential of −100 mV to −20 mV at 30 s intervals. The effects of 6 min capsaicin (30 µM) application and 8 min recovery on superimposed traces of *I*_Na_ are shown in Figure 1A. The bath application of capsaicin caused a gradually progressing suppression of *I*_Na_, which reached steady-state levels within 4–5 min (Figure 1B). The recovery was partial during the experiments lasting up to 15 to 18 min. The current–voltage (*I*-*V*) relationship for *I*_Na_ is presented in Figure 1C. No significant changes in the *I*-*V* relationship were observed; the threshold, peak, and reversal potentials for *I*_Na_ remained unaltered (n = 5). The extent of the capsaicin suppression of *I*_Na_ was not altered by changes in the test potentials (Figure 1D). The concentration-dependency of capsaicin inhibition is presented in Figure 1E. The IC_50_ value and apparent Hill coefficient were 42.7 µM and 1.4, respectively (*n* = 4–7). The bath application of capsazepine (3 µM) for 10 min did not alter the peak *I*_Na_ (*n* = 4). The extent of capsaicin inhibition of *I*_Na_ was not altered by the co-application of 3 µM capsazepine, an antagonist of the TRPV1 receptor, and 30 µM capsaicin (*n* = 6–7, *p* > 0.05; *t*-test).

### 2.2. The Effects of Capsaicin on L-Type Ca^2+^ Channels (I_L-Ca_)

*I*_L-Ca_ was studied in the presence of extracellular TEA^+^ and intracellular Cs^+^ to suppress K^+^ currents. In addition, the contaminating Na^+^ current was eliminated by applying test potentials from a relatively depolarized potential of −50 mV to induce the inactivation of *I*_Na_ [20]. The superimposed traces of the currents elicited by the test potentials from −50 mV to +20 mV in the control, after a 5 min application of 30 µM capsaicin and 7 min recovery, are shown in Figure 2A. The bath application of capsaicin caused a steadily progressing inhibition of *I*_L-Ca_, which was detectable within 30 sec and reached a steady-state level within 3–5 min, and the recovery was partial (Figure 2B). The *I-V* relationships in the control and in the presence of 30 µM capsaicin are presented in Figure 2C (*n* = 5). The relationship between the test potentials and the extent of capsaicin inhibition of *I*_L-Ca_ was presented in Figure 2D (*n* = 5). Capsaicin inhibited *I*_L-Ca_ in a concentration-dependent manner with an IC_50_ value of 34.9 µM and an apparent Hill coefficient of 1.3 (*n* = 3–6; Figure 2E). The application of capsazepine (3 µM) for 10 min did not affect the peak *I*_Na_ (*n* = 4). The extent of capsaicin inhibition of *I*_Na_ was not altered by the co-application of 3 µM capsazepine, an antagonist of the TRPV1 receptor, and 30 µM capsaicin (*n* = 5–7, *p* > 0.05; *t*-test).

### 2.3. The Effects of Capsaicin on the Delayed Rectifier K^+^ Channels (I_K_)

The delayed rectifier K^+^ current in mammalian ventricular myocytes consists of a rapidly activating but inwardly rectifying *I*_Kr_ and a slowly activating I_Ks_ [21,22].

Rapidly activating delayed rectifier (*I*_Kr_) was quantified by applying a test pulse from −40 mV to +40 mV and measuring the deactivating tail currents in response to repolarizing test pulse of −40 mV. The difference between the peak of the tail current and the current at the beginning of test potential was taken as an estimate of *I*_Kr_. The contribution of *I*_Ks_ was suppressed by 3 µM of selective *I*_Ks_ blocker HMR 1556 to increase the proportion of *I*_Kr_. Traces of tail currents elicited by test potentials from +40 mV to −40 mV are presented for control and after 6 min application of 10 µM capsaicin and 7 min recovery in Figure 3A. The bath application of 10 µM capsaicin caused the inhibition of *I*_Kr_, which was evident at 30 sec and reached a steady-state level within 5 min, and recovery was almost complete within the time course of the experiment (Figure 3B). The effects of 3 µM capsaicin on the *I-V* relationship of *I*_Kr_ are presented in Figure 3C (*n* = 5). The extent of the capsaicin inhibition at test potentials ranging from −20 to 50 mV is shown in Figure 3D (*n* = 5).

The concentration-dependent effects of capsaicin are shown in Figure 3E. The IC_50_ value and apparent Hill coefficient were 3.4 µM and 1.4, respectively (*n* = 4–8). The bath application of 3 µM capsazepine did not affect the peak amplitudes of *I*_Kr_ tails (*n* = 4). The extent of the capsaicin inhibition of *I*_Kr_ was not altered by the co-application of 3 µM capsazepine and capsaicin (3 µM) (Figure 3F; *n* = 5–6, *p* > 0.05).

Slowly activating delayed rectifier (*I*_Ks_) was measured as the time-dependent current induced by 500 ms pulses from −50 to +60 mV. The superimposed current traces are shown for the control after an 8 min application of 10 µM capsaicin and a 5 min recovery in Figure 4A. The application of capsaicin caused a gradually developing inhibition of *I*_Ks_, which reached a steady-state level within 6 min and recovered partially during the experiments (Figure 4B). Figure 4C shows the effect of 10 µM capsaicin on the *I-V* relationship of *I*_Ks_ (*n* = 5). The extent of capsaicin inhibition at test potentials ranging from −40 to 60 mV is shown in Figure 4D (*n* = 5). The concentration-dependent inhibition of *I*_Ks_ is presented in Figure 4E. The IC_50_ value and apparent Hill coefficient were 14.7 µM and 1.3, respectively (*n* = 4–6). The bath application of 3 µM capsazepine did not affect the peak amplitudes of *I*_Ks_ (*n* = 4). The extent of capsaicin inhibition of *I*_Ks_ was not altered by the co-application of 3 µM capsazepine and 3 µM capsaicin (Figure 4F; *n* = 5–6, *p* > 0.05).

### 2.4. The Effect of Capsaicin on Transient Outward K^+^ Current (I_to_)

*I*_to_ was activated by 400 ms test pulses from −80 mV to +60 applied at 30 s intervals and defined as the difference between the initial transient peak of the current and the maintained current at the end of the pulse. The superimposed current traces in the absence and 6 min presence of 10 µM capsaicin and during an 8 min recovery are presented in Figure 5A. The time course of the effect of bath application of capsaicin on the peak amplitudes of *I*_to_ is shown in Figure 5B. The *I-V* relationships of *I*_to_ in the absence and presence of 10 µM capsaicin are shown in Figure 5C (*n* = 7). The relationship between the test potentials and the extent of the capsaicin inhibition of *I*_to_ was presented in Figure 5D (*n* = 7). Figure 5E shows the concentration–inhibition curve of capsaicin on *I*_to_ (*n* = 4–7 cells). The IC_50_ and apparent Hill coefficient were 9.6 µM and 1.4, respectively. Capsazepine (3 µM) alone did not affect the peak amplitudes of *I*_to_ (*n* = 4). The extent of capsaicin inhibition of *I*_to_ was not altered by the co-application of 3 µM capsazepine and 10 µM capsaicin (Figure 5F; *p* > 0.05 *t*-test).

### 2.5. The Effect of Capsaicin on the Inward Rectifier K^+^ Current (I_K1_)

*I*_K1_ was activated from the holding potential of −80 mV to the test potential of −120 mV. The amplitude of *I*_K1_ was determined by measuring the peak current relative to zero current. The traces of the currents in the control, in the presence of 30 µM capsaicin, and during recovery are shown in Figure 6A. Figure 6B shows the time course of the capsaicin effect and washout on *I*_K1_. The *I-V* relationships of *I*_K1_ in the absence and presence of 30 µM capsaicin are presented in Figure 6C (*n* = 5). The relationship between the test potentials and the extent of the capsaicin inhibition of *I*_K1_ is presented in Figure 6D (*n* = 5). Figure 6E demonstrates the concentration–inhibition curve of capsaicin on *I*_K1_ (*n* = 4–6 cells). The IC_50_ and apparent Hill coefficient were 38.8 µM and 1.4, respectively. Capsazepine (3 µM) alone did not affect the peak amplitudes of *I*_K1_. The extent of capsaicin inhibition of *I*_K1_ was not altered by the co-application of 3 µM capsazepine and 10 µM capsaicin (Figure 6F; *p* > 0.05; *t*-test).

## 3. Discussion

The results have demonstrated that capsaicin at concentrations higher than those required for the activation of TRPV1 channels directly inhibits the functions of multiple ion channels in rabbit ventricular myocytes. Capsaicin appears to effectively inhibit delayed rectifier K^+^ currents (*I*_Kr_ and *I*_Ks_) and *I*_to_, with IC_50_ values of 3.4 µM,14.7 µM, and 9.6 µM, respectively. On the other hand, *I*_Na_, *I*_L-Ca_, and *I*_K1_ seem to be significantly less sensitive to capsaicin, with respective IC_50_ values of 42.7 µM, 34.9 µM, and 38.8 µM.

Although this is the first report on the capsaicin inhibition of *I*_Kr_ and *I*_Ks_, in earlier studies, capsaicin was reported to inhibit *I*_to_, and the *I*_K1_, with IC_50_ values of 6.4 µM and 46.9 µM, respectively, in rat ventricular myocytes [23]. Similarly, the inhibition of *I*_to_ by capsaicin (IC_50_ = 5 µM) was reported in rat atrial myocytes [24]. In addition, capsaicin suppresses *I*_HERG_, which corresponds to a rapid component of the delayed rectifier K^+^ channel, expressed in *Xenopus* oocytes with an IC_50_ of 17.5 µM [25]. These results are in agreement with our findings on K^+^ currents in rabbit ventricular myocytes.

Capsaicin, at significantly higher concentrations, suppressed *I*_Na_, *I*_L-Ca_, *I*_K1_ with IC_50_ values of 42.7 µM, 34.9 µM, and 38.8 µM, respectively. In an earlier study, capsaicin, at lower concentrations (0.4–4 µM), was reported to significantly suppress *I*_Na_ in rat atrial myocytes [26]. Similarly, capsaicin, at a significantly lower concentration of 0.33 µM, inhibited Vmax during phase 0 of action potential in guinea-pig papillary muscle [10]. However, other studies reported that capsaicin suppresses Vmax of action potentials in the concentration range of 30–120 µM in guinea-pig papillary muscle [27]. In a recent study, in agreement with our findings (IC_50_ = 42.7 µM), capsaicin inhibited cardiac Na^+^ channels (Na_V_1.5) with an IC_50_ of 60.2 µM in HEK-293 cells [28]. Although the present study is the first to report the inhibition (IC_50_ = 34.9 µM) of *I*_L-Ca_ by capsaicin in cardiomyocytes, in agreement with our findings, capsaicin, at similar concentrations (10–30 µM) suppressed Vmax of Ca^2+^-dependent action potentials in rabbit atrioventricular node [29] and sinoatrial cells [30].

The simulation with LabHEART, a computer model of rabbit ventricular action potentials [31], revealed that the suppression of *I*_Ks_, *I*_Kr_, and *I*_to_ at a concentration of 10 µM, capsaicin, as expected, causes a marked prolongation (39%) of action potential duration at 50% repolarization (APD_50_), whereas the addition of the inhibition of *I*_Na_, *I*_L-Ca_, and *I*_K1_ at higher capsaicin concentration (30 µM) results in a 27% shortening of APD_50_ (Figure 7A). In earlier studies, capsaicin at concentrations higher than 1 µM has been reported to prolong APD in rat ventricular [23,24] and guinea-pig atrial [19] myocytes. On the other hand, capsaicin, at concentrations of 10 µM and higher, has been shown to decrease APD in guinea-pig papillary muscles [10,27,32] and canine cardiomyocytes for zucapsaicin [33]. Both the increase and decrease in APD can potentially have arrhythmogenic effects depending mainly on the underlying pathophysiological mechanisms, such as ischemic heart diseases. Our LabHEART simulation of rabbit ventricular myocytes also indicates that the high concentration of capsaicin (30 µM) can markedly depress intracellular Ca^2+^ transients (Figure 7B) and tension development (Figure 7C). 

TRPV1 channels are not expressed in adult rat [34] and mouse [35,36] cardiomyocytes but may play roles at earlier developmental stages of myocytes [36,37]. In line with these findings, the effects of capsaicin on ion channels tested in this study (*I*_Kr_, *I*_Ks_, *I*_to_) were not reversed by capsazepine, a TRPV1 antagonist. In addition, the bath application of capsaicin (1–30 µM), even at concentrations 10–100 times higher than those required to activate TRPVI channels, did not change the holding currents under patch-clamp conditions to record *I*_Na_ and *I*_L-Ca_ (*n* = 14, data not shown), suggesting that TRPV1 is not involved in the observed effects of capsaicin. Cannabinoid receptors (CB1 and CB2) have been shown to be expressed in rat hearts [38], and capsaicin at concentrations higher than 10 µM was reported to bind to CB1 and CB2 receptors [39]. However, the co-application of specific CB1 antagonist SR141716A (2 µM) or specific CB2 antagonist SR 144528 (2 µM) and capsaicin did not alter the extent of the capsaicin inhibition of *I*_Na_ and *I*_L-Ca_ (Paired *t*-test; *p* > 0.05; *n* = 4–5), suggesting that CB1 and CB2 receptors are not involved in the capsaicin inhibition of these ion channels.

It is likely that capsaicin, a highly lipophilic agent with a LogP (octanol–water partition coefficient) value of 3.8, permeates the lipid membrane and then diffuses into the lipid membrane, and then alters the functional properties of the ion channels. Molecular dynamics simulations suggested that capsaicin is localized to the bilayer/solution interface [40]. Capsaicin was reported to regulate the functions of voltage-gated Na^+^ and K^+^ channels, as well as antibiotic-induced ion channels [41], by altering the biophysical properties of the lipid membranes, such as the bilayer elasticity [40,42], dipole potential [43], and lipid disordering [44].

The effect of capsaicin on the ion channels tested in this study reached a maximal level within several minutes (3–6 min) of application. Similarly, the actions of several lipophilic modulators, including capsaicin [42,45,46], endocannabinoids [47,48,49], and general anesthetics [50,51] and steroids [52] on various ion channels require 5–15 min to reach their maxima with, in many cases, Hill coefficients of higher than unity in their concentration–effect curves, suggesting that the multiple binding sites for these allosteric modifiers are located inside the lipid membrane and require a relatively slow (in minutes) time course to reach equilibrium to modulate the functions of these channels. Thus, the accumulation of capsaicin in the lipid membrane appears to be a function of both the exposure time and concentration of capsaicin. Thus, while low capsaicin concentrations require longer exposure times, higher concentrations reach a threshold concentration at faster time scales to affect channel functions. In line with this assumption, the application of low concentrations (0.1–0.3 µM) for relatively long exposures (10–20 min) causes gradually developing suppression of contractility in guinea-pig papillary [10] and ventricular [19] muscle and whole heart [11] preparations. In another in vitro study, the 30 min administration of capsaicin, at concentrations as low as 1 nM, significantly inhibited the contractions of rat ventricular papillary muscle in a reversible manner [17]. In agreement with these findings, we have found that time to reach steady-state inhibition by capsaicin was inversely correlated with capsaicin concentrations applied to the ion currents investigated in this study (Figure 8). In addition to partitioning into the lipid bilayer and subsequently altering the biophysical characteristics of the plasma membrane, capsaicin can bind directly to ion channel domains embedded in cell membranes and affect the energy requirements for the gating-related conformational changes in ion channels [49].

In recent years, capsaicin has been administered as a dietary supplement for weight management and appetite suppression. Pharmacokinetic studies indicate that about 80% of ingested capsaicin is absorbed through the gastrointestinal system, leading to peak concentrations at sub-µM levels with a half-time of 30–60 min [1,53]. Thus, the regular intake of low-dosage capsaicin supplements (1–30 mg/day) would not reach blood levels that would cause the significant inhibition of K^+^ channels investigated in this study. However, as mentioned earlier, capsaicin is a highly lipophilic compound (LogP = 3.8), and its concentration in the cell membrane is expected to be considerably higher than blood levels. In animal studies, after intravenous or subcutaneous administrations, the capsaicin concentrations in the brain and spinal cord were found to be five-fold higher than that in the blood [54,55]. Thus, it is possible that chronic and frequent intake of high-dosage ( >100 mg/day) capsaicin may lead to significant capsaicin partitioning into lipid membranes and trigger cardiac arrhythmias, especially in patients with comorbid ischemic heart diseases.

## 4. Materials and Methods

Male New Zealand white rabbits (ca. 2–2.5 kg) were anesthetized by sodium pentobarbitone (40 mg/kg) injection into a marginal ear vein, and following anesthesia, the animal was killed by the removal of the heart. This study was carried out in accordance with the recommendations in the Guide for the Care and Use of Laboratory Animals of the National Institutes of Health and approved by the Animal Care Committee of Bogomoletz Institute of Physiology of the National Academy of Science of Ukraine (approval code: 345/5.7WK). Ventricular myocytes were isolated according to minor modifications of the method described earlier [56,57]. Briefly, the hearts were retrogradely mounted and perfused according to the Langendorff method at a constant flow of 10 mL g heart^−1^ min^−1^ and at 37 °C with a cell isolation solution containing (mM): 130 NaCl, 5.4 KCl, 1.4 MgCl_2_, 0.75 CaCl_2_, 0.4 NaH_2_PO_4_, 5 HEPES, 10 glucose, 20 taurine, and 10 creatine set to pH 7.3 with NaOH. After the stabilization of the heart contractions, perfusion was continued for 4 min with Ca^2+^-free isolation solution containing 0.1 mM EGTA, and then for 6 min with cell isolation solution containing 0.05 mM Ca^2+^-, 0.8 mg/mL collagenase (type 1; Worthington Biochemical Corp, USA) and 0.075 mg/mL protease (type X1 V; Sigma, Taufkirchen, Germany). Following enzyme treatment, the heart was removed from the Langendorff perfusion system, and the ventricles were excised, minced, and gently shaken in a collagenase-containing isolation solution supplemented with 1 % BSA. The cells were filtered from this solution at 4 min intervals and resuspended in an isolation solution containing 0.75 mM Ca^2+^. The shaking and filtration process was repeated up to five times.

### 4.1. Whole Cell Patch-Clamp Technique

Myocytes were dispersed and allowed to settle for at least one hour at room temperature (22–24 °C) prior to their use. The measurements were performed only in quiescent myocytes displaying normal morphology and clear striated appearance. The whole-cell patch-clamp technique was used to evaluate individual ionic currents using an Axopatch 200B amplifier (Molecular Devices, Sunnyvale, CA, USA) linked to an A/D interface (Digidata 1322; Molecular Devices). The analog signal was filtered using a four-pole Bessel filter with a bandwidth of 5 kHz and digitized at a sampling rate of 10 kHz under software control (PClamp 10.6.2.2, Molecular Devices, Sunnyvale, CA, USA). Heat-polished borosilicate glass pipettes (World Precision Instruments, Sarasota, FL, USA) with a tip resistance of 1 to 2 MΩ were used to establish GΩ seals and continuity with the intracellular medium. The cell capacitance (C_m_) was calculated by integrating the area under an uncompensated capacity transient elicited by a 10-mV depolarizing pulse from a holding potential of −80 mV. The total series resistance (R_s_) between the pipette interior and the cell membrane in the whole-cell configuration was calculated from the estimates of C_m_ (172 ± 14 pF, *n* = 247 from 36 rabbits) and the time constant (τ_c_) of the capacitative current decay from the equation τ_c_ = R_s_ × C_m_. The mean R_s_ for the pathway between the pipette and the cell membrane after the rupture of the membrane seal were calculated to be 3.87 ± 0.32 MΩ (*n* = 247). After the establishment of the whole-cell configuration and the measurement of C_m_, the R_s_ were compensated to > 60%. The junction potentials under our conditions were approximately −3 mV and were not corrected.

The control perfusate was a modified Tyrode solution containing (in mM): 137 NaCl, 5.4 KCl, 1 MgCl_2_, 2 CaCl_2_, 10 HEPES, 10 glucose; titrated with NaOH to pH 7.4. Extracellular solution for recordings of Na^+^ currents consisted of (in mM): 100 TEACl, 40 NaCl, 10 glucose, 1 MgCl_2_, 5 CsCl, 0.1 CaCl_2_, 10 HEPES [adjusted to pH 7.4 with CsOH; [20]], and 10 µM nifedipine included to suppress L-type Ca^2+^ current. Intracellular solution contained (in mM) 135 CsCl, 5 NaCl, 10, EGTA, 10 HEPES, and 1 MgATP (adjusted to pH 7.25 with CsOH). For the recording of Ca^2+^ currents, the whole-cell bath solution contained (in mM): 95 NaCl, 50 TEACl, 2 MgCl_2_, 2 CaCl_2_, 10 HEPES, and 10 glucose (adjusted to pH 7.35 with NaOH). The pipette solution contained (in mM): 140 CsCl, 10 TEACl, 2 MgCl_2_, 2 HEPES 1 MgATP and 10 EGTA [adjusted to pH 7.25 with CsOH [58]. For recording K^+^ currents, the external solution contained (in mM): 132 NaCl, 4 KCl, 1.8 CaCl_2_, 1.2 MgCl_2_, 0.2 BaCl_2_, 10 glucose, 10 HEPES, 5 4-aminopyridine (4-AP), 0.01 nifedipine (pH adjusted to 7.4 with NaOH). The pipette solution for *I*_K_ recordings contained (in mM) 120 potassium glutamate, 20 KCl, 2 MgCl_2_, 10 HEPES, and 5 Mg-ATP; adjusted to pH 7.2 with KOH. When *I*_Kr_ was recorded, *I*_Ks_ was inhibited by using 3 µM selective *I*_Ks_ blocker HMR 1556 (Tocris, Minneapolis, MN, USA). During *I*_Ks_ measurements, *I*_Kr_ was blocked by 3 µM dofetilide (Sigma, St. Louis, MO, USA), and the bath solution contained 0.1 µM forskolin (Sigma, St. Louis, MO, USA). Under these conditions, dofetilide (3 µM) and HMR 1556 (3 µM) alone effectively (85–95%) blocked *I*_Kr_ and *I*_Ks_, respectively (*n* = 4–5). In these recordings, *I*_K1_, *I*_to_, and *I*_L-Ca_ were blocked by 200 µM BaCl_2_, 5 mM 4-AP, and 10 µM nifedipine, respectively. For recordings of *I*_to_, 10 µM nifedipine, 20 µM tetrodotoxin (TTX), and 3 µM dofetilide were included to eliminate *I*_L-Ca_, *I*_Na_, and *I*_Kr_, respectively. These blockers alone (BaCl_2_, 4-AP, TTX, and nifedipine effectively blocked (85–95%; *n* = 3–4; without leak subtraction) *I*_K1_, *I*_to_, *I*_Na_, and *I*_L-Ca_, respectively.

The experiments were performed at room temperature (22–24 °C). The changes of the external solutions and the application of drugs were performed using a multi-line perfusion system with a common outflow connected to the recording chamber. A perfusion rate of 2 mL/min was used routinely in a recording chamber with a volume of 200 µL.

Capsaicin was from Sigma (St. Louis, MO, USA). It was dissolved in 100 % DMSO, and final concentrations were diluted from stock solutions. The stocks were kept at −20 °C until their use. The highest final concentration of DMSO in the extracellular solutions (0.03% *v*/*v*) did not affect the amplitudes of the membrane currents investigated in this study. Capsazepine (dissolved in DMSO), other reagents, and chemicals used in our experiments were purchased from Sigma-Aldrich (St. Louis, MO, USA).

### 4.2. Statistical Analysis

All of the cumulative results are expressed as mean ± SE or SEM as indicated.

Statistical significance among groups was determined using *t*-test, ANOVA, or pair-wise comparisons (Mann–Whitney U-Test) followed by Bonferroni Post-hoc analysis. Statistical analysis of the data was performed using Origin 7.0 software (OriginLab Corp., Northampton, MA, USA). *p* < 0.05 was considered statistically significant. The concentration–response parameters were obtained by fitting the data to the logistic equation in Origin 7.0 software,
y = E_max_/(1 + [x/EC_50_]^n^),
where x and y are the concentration and response, respectively, E_max_ is the maximal response, EC_50_ is the half-maximal concentration, and n is the slope factor (apparent Hill coefficient).

## Figures and Tables

**Figure 1 pharmaceuticals-15-01187-f001:**
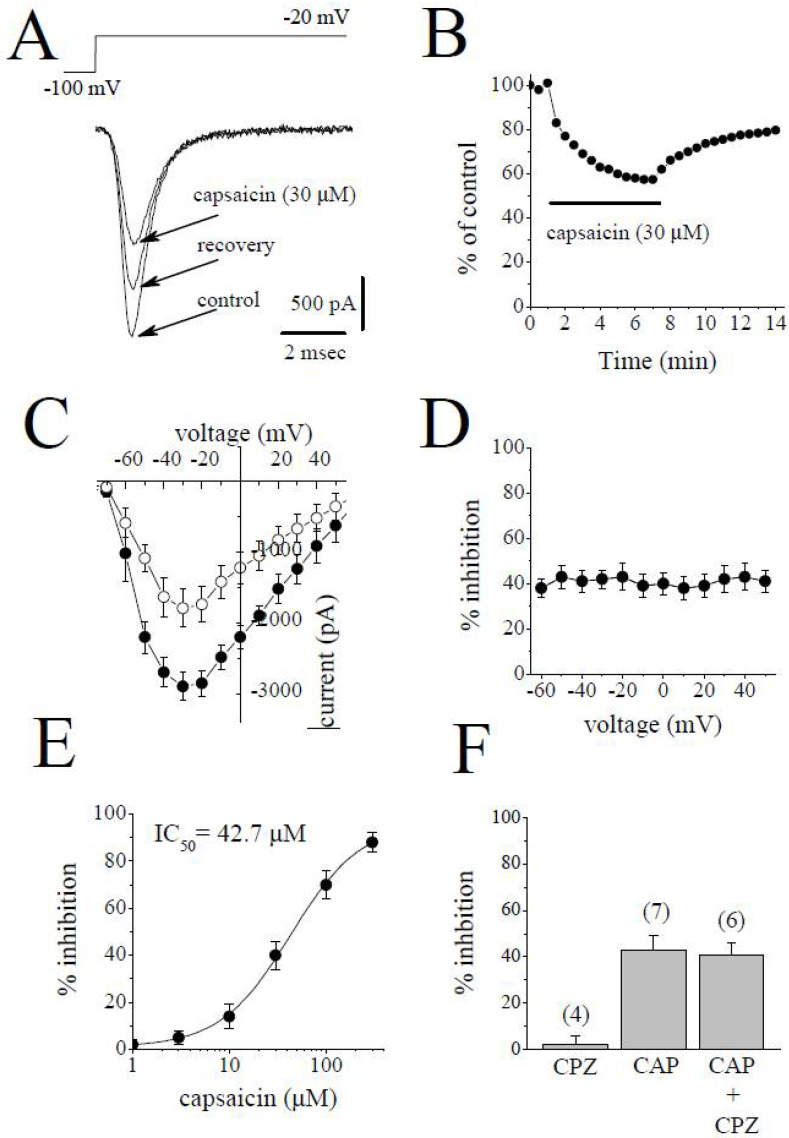
Capsaicin inhibits *I*_Na_ in rabbit ventricular myocytes. (**A**) Superimposed current traces in control conditions, 6 min after exposure to 30 μM capsaicin, and 8 min recovery. The pulse protocol to activate *I*_Na_ is shown as in inset. (**B**) Time course of the effect of capsaicin on the maximal amplitudes of *I*_Na_ as % of control current calculated as the mean of three consecutive control currents at the beginning of the experiment. (**C**) Current–voltage relationship of *I*_Na_ in the absence and presence of 30 μM capsaicin are presented with filled and open circles, respectively (*n* = 5). (**D**) Relationship between the extent of capsaicin (30 µM) inhibition of *I*_Na_ and test potential (*n* = 5; *p* > 0.05, ANOVA). (**E**) Concentration-inhibition curve for capsaicin inhibition of *I*_Na_ (*n* = 4–7). (**F**) The effect of capsazepine (3 µM) and the extent of capsaicin (30 µM) inhibition of *I*_Na_ in the absence and presence of 3 µM capsazepine. The number of cells tested for each group was presented on top of each bar (*p* > 0.05; *t*-test).

**Figure 2 pharmaceuticals-15-01187-f002:**
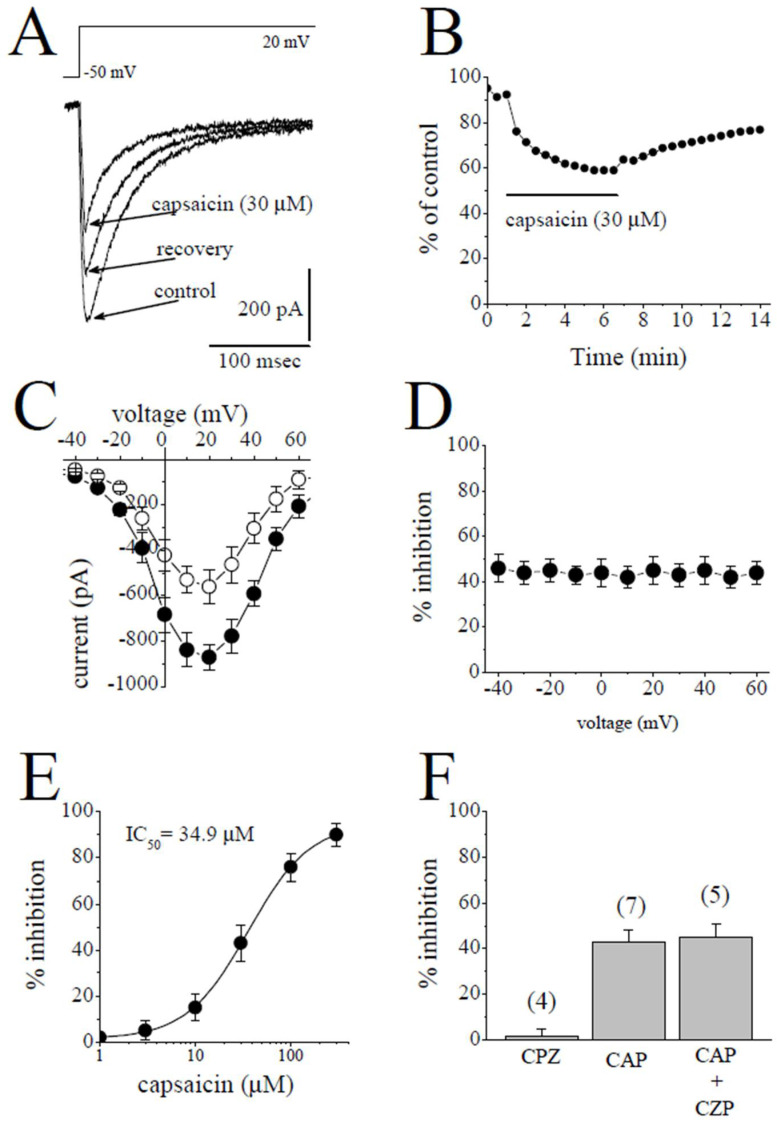
Capsaicin suppresses the *I*_L-Ca_ in rabbit ventricular myocytes. (**A**) Superimposed current traces in control, after 5 min exposure to 30 μM capsaicin and recovery. The pulse protocol to evoke *I*_L-Ca_ is shown as an inset. (**B**) Time course of the effect of capsaicin on the maximal amplitudes of *I*_L-Ca_ presented as % of control calculated from the mean of three control currents. (**C**) Current–voltage curves of *I*_L-Ca_ in the absence and presence of 30 μM capsaicin are presented with filled and open circles, respectively (*n* = 5). (**D**) The relationship between test potential and the capsaicin (30 µM) inhibition of *I*_L-Ca_ (*n* = 5; *p* > 0.05, ANOVA). (**E**) Concentration-inhibition relationship for capsaicin suppression of *I*_L-Ca_ (*n* = 3–6). (**F**) The effect of capsazepine (3 µM) and the extent of capsaicin (30 µM) inhibition of *I*_L-Ca_ in the absence and presence of 3 µM capsazepine. The number of cells tested for each group was presented on top of each bar (*p* > 0.05; *t*-test).

**Figure 3 pharmaceuticals-15-01187-f003:**
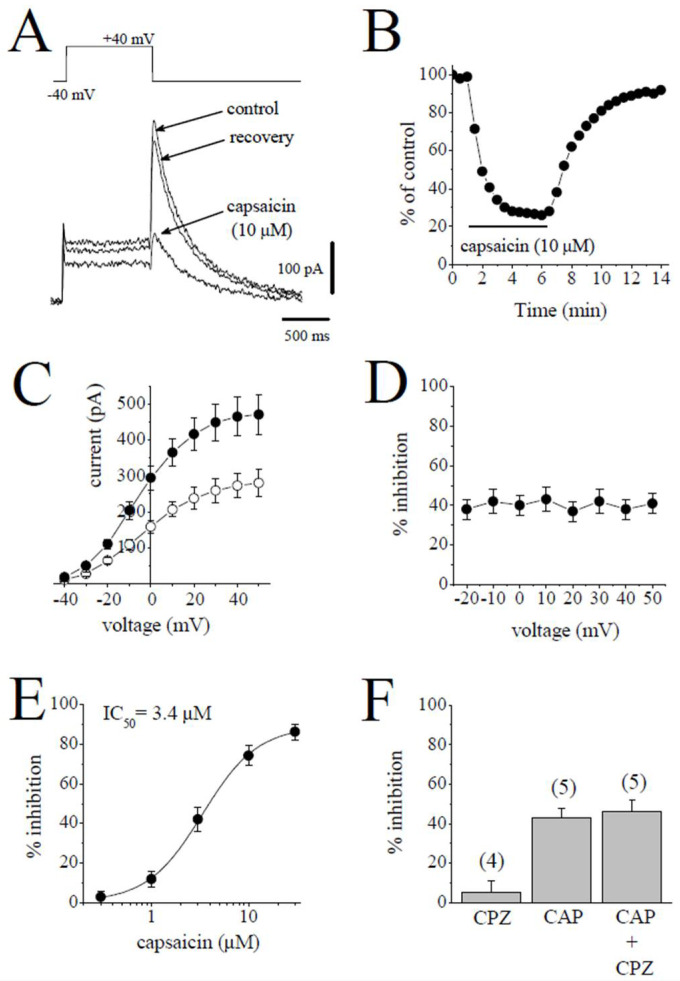
Capsaicin suppresses the *I*_Kr_ tail current. (**A**). Superimposed current traces in control, after 6 min exposure to 10 μM capsaicin and recovery. *I*_Kr_ tails were measured as a time-dependent component of the tail current activated in response to membrane repolarization. The pulse protocol to activate *I*_Kr_ is presented as an inset. (**B**) Time course of the effect of capsaicin effect on the maximal amplitudes of *I*_Kr_ and washout presented as % of control calculated from the mean of three control currents (*n* = 5). (**C**) Current–voltage relationships of *I*_Kr_ in absence (filled circles) and presence (open circles) of 3 µM capsaicin are shown (*n* = 6). (**D**) The relationship between test potential and the capsaicin (3 µM) inhibition of *I*_Kr_ (*n* = 6; *p* > 0.05, ANOVA). (**E**) Concentration-inhibition curve for capsaicin inhibition of *I*_Kr_ tails (*n* = 4–8). (**F**) The effect of capsazepine (3 µM) alone and the extent of capsaicin (3 µM) inhibition of *I*_Kr_ in the absence and presence of 3 µM capsazepine. The number of cells tested for each group was presented on top of each bar (*p* > 0.05; *t*-test).

**Figure 4 pharmaceuticals-15-01187-f004:**
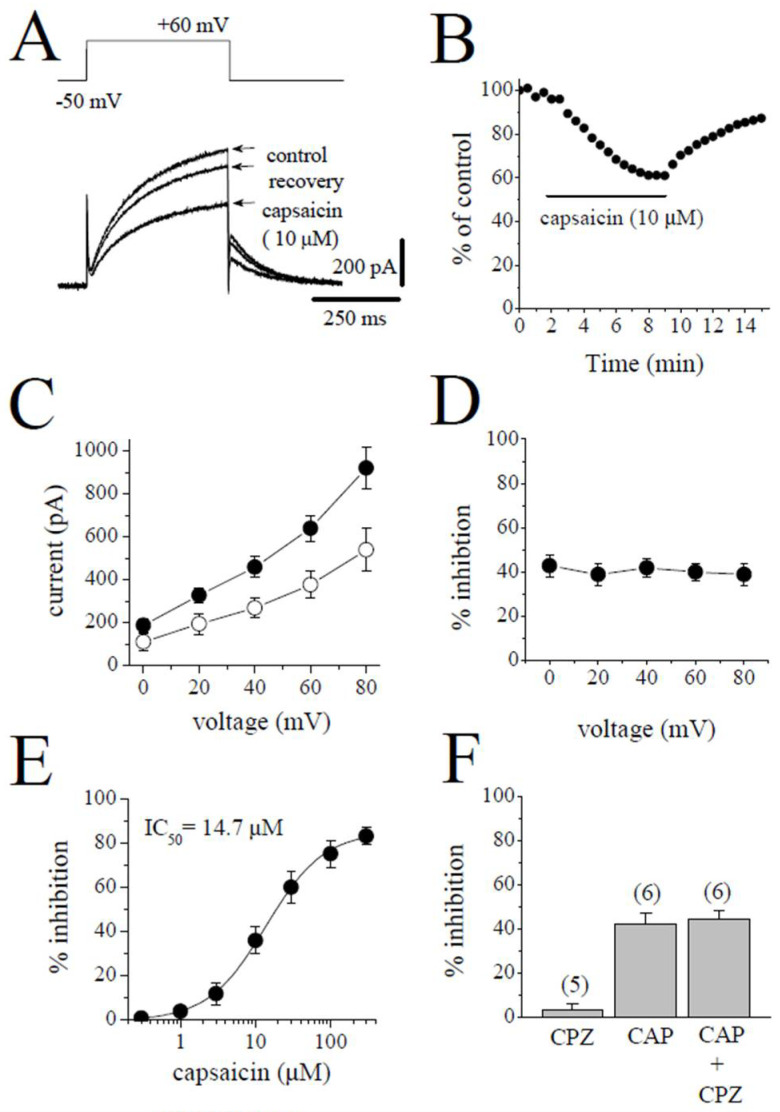
Capsaicin suppresses the *I*_Ks_ component of the delayed rectifying *I*_K_. (**A**) Superimposed current traces in control, after 8 min exposure to 10 μM capsaicin, and recovery. The pulse protocol to activate *I*_Ks_ is shown as an inset. (**B**) Time course of the effect of capsaicin effect on the maximal amplitudes of *I*_Ks_ and washout presented as % of control calculated from the mean of three control currents. (**C**) Current–voltage relationships of I_Ks_ in the absence and presence of 10 μM capsaicin are presented with filled and open circles, respectively (*n* = 5). (**D**) The relationship between test potential and the capsaicin (10 µM) inhibition of *I*_Ks_ (*n* = 5; *p* > 0.05, ANOVA). (**E**) Concentration-inhibition curve for capsaicin inhibition of *I*_Ks_ (*n* = 4–6). (**F**) The effect of capsazepine (3 µM) and the extent of capsaicin (10 µM) inhibition of *I*_Ks_ in the absence and presence of 3 µM capsazepine. The number of cells tested for each group was presented on top of each bar (*p* > 0.05; *t*-test).

**Figure 5 pharmaceuticals-15-01187-f005:**
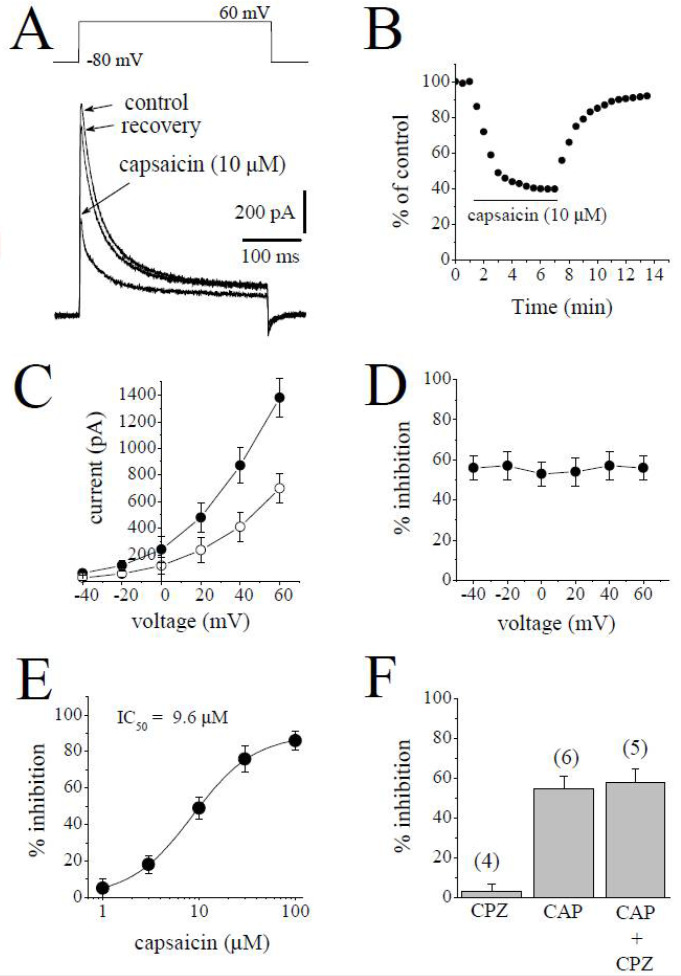
The effects of capsaicin on the transient outward *I*_to_ in rabbit ventricular myocytes. (**A**) Traces of *I*_to_ obtained in response to depolarizations from a holding potential of −80 mV to +60 mV are presented in control after 6 min exposure to 10 μM capsaicin and recovery. The pulse protocol to activate *I*_to_ is presented as an inset. (**B**) Time course of the effect of capsaicin effect on the maximal amplitudes of *I*_to_ and washout presented as % of control calculated from the mean of three control currents. (**C**) Current–voltage relationships of *I*_to_ in absence (filled circles) and presence (open circles) of 10 µM capsaicin are shown (*n* = 5). (**D**) The relationship between test potential and the capsaicin (10 µM) inhibition of *I*_to_ (*n* = 5; *p* > 0.05, ANOVA). (**E**) Concentration–inhibition of capsaicin for *I*_to_ (*n* = 4–7). (**F**) The effect of capsazepine (3 µM) and the extent of capsaicin (10 µM) inhibition of *I*_to_ in the absence and presence of 3 µM capsazepine. The number of cells tested for each group was presented on top of each bar (*p* > 0.05; *t*-test).

**Figure 6 pharmaceuticals-15-01187-f006:**
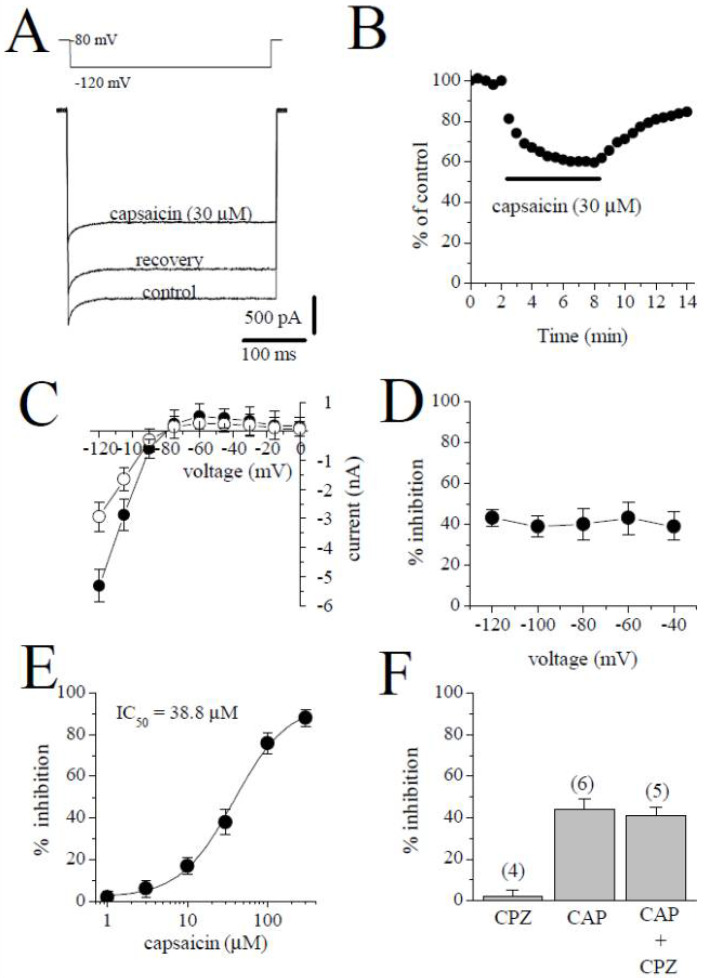
The effect of capsaicin on *I*_K1_. (**A**) Traces of *I*_K1_ obtained in control after 6 min exposure to 30 μM capsaicin and recovery. The pulse protocol to activate *I*_K1_ is presented as an inset. (**B**) Time course of the effect of capsaicin effect on *I*_K1_ and washout presented as % of control calculated from the mean of three control currents. (**C**) Current–voltage relationships of I_K1_ in the absence and presence of 30 μM capsaicin were presented with filled and open circles, respectively (*n* = 5). (**D**) The relationship between test potential and the capsaicin (10 µM) inhibition of *I*_K1_ (*n* = 5; *p* > 0.05, ANOVA). (**E**) The effect of increasing concentration of capsaicin on *I*_K1_ (*n* = 4–6). (**F**) The effect of capsazepine (3 µM) and the extent of capsaicin (30 µM) inhibition of *I*_K1_ in the absence and presence of 3 µM capsazepine. The number of cells tested for each group was presented on top of each bar (*p* > 0.05; *t*-test).

**Figure 7 pharmaceuticals-15-01187-f007:**
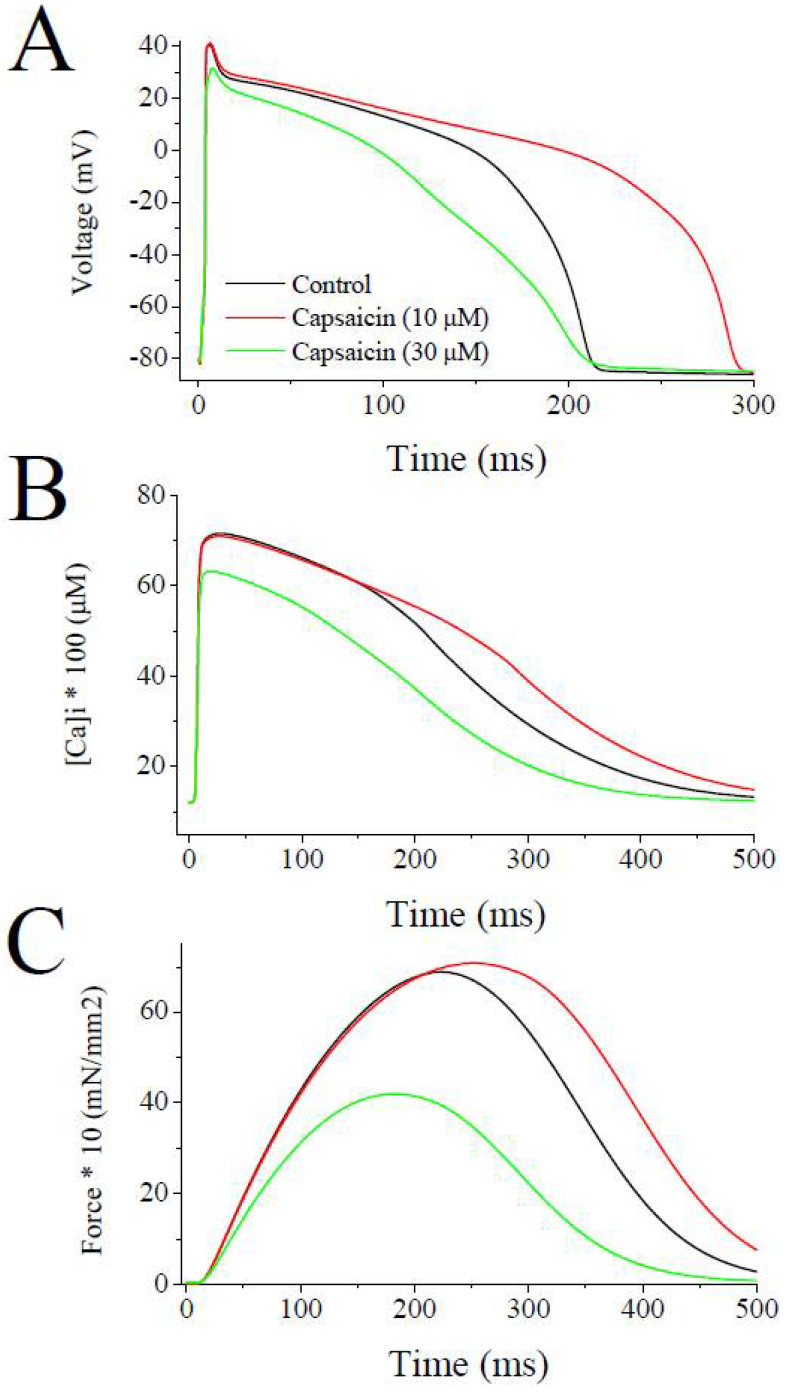
Effects of low and high concentrations of capsaicin on the simulated action potentials, intracellular Ca^2+^ transients, and force development during a single twitch response using LabHEART simulations. (**A**) At 10 µM capsaicin concentrations, the inhibition of *I*_Kr_, *I*_Ks_, *I*_to_ was incorporated. Whereas, at 30 µM capsaicin concentrations inhibitions of *I*_Kr_, *I*_Ks_, *I*_to_, *I*_K1_, *I*_Na_, and *I*_Ca-L_ were incorporated into the model. At 10 µM concentration, capsaicin induced marked prolongation (39%) of APD_50_, while at higher concentration (30 µM) significantly shortened (27%) the APD_50_. (**B**) Capsaicin, at the concentration of 10 µM, caused a slight delay in decay phase of Ca^2+^ transient. At higher concentration of 30 µM, capsaicin induced a marked suppression of Ca^2+^ transient. (**C**) Capsaicin, at 10 µM concentrations, caused a slight delay in decay phase of force development, but at higher concentrations of 30 µM, capsaicin induced a marked inhibition of force development during a simulated twitch response of a rabbit ventricular cardiomyocyte.

**Figure 8 pharmaceuticals-15-01187-f008:**
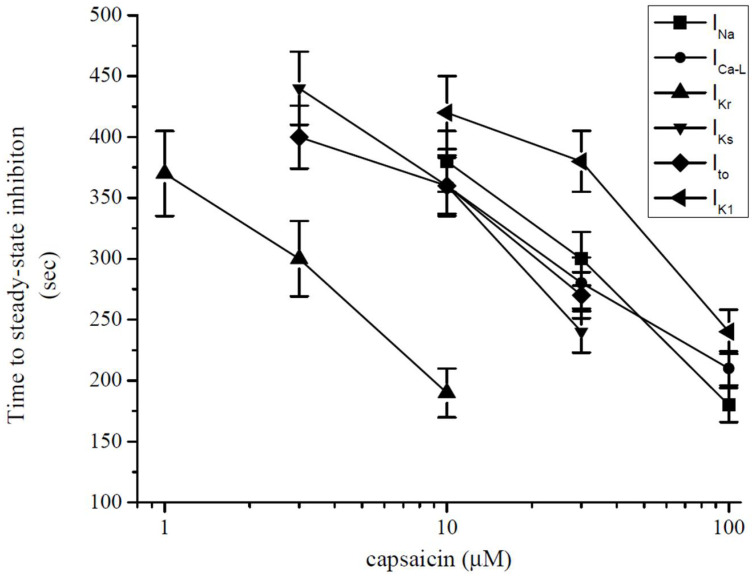
Relationship between time to reach steady-state maximal inhibition and capsaicin concentration for *I*_Na_, *I*_L-Ca_, *I*_Kr_, *I*_Ks_, *I*_to_, and *I*_K1_ (each data point indicates mean ± SEM from 5–7 cells). Steady-state level was determined when three consecutive data points had the same (within the 97%) values.

## Data Availability

Data are contained within the article.

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
