# Peer review of "Capsaicin Inhibits Multiple Voltage-Gated Ion Channels in Rabbit Ventricular Cardiomyocytes in TRPV1-Independent Manner"

_pharmaceuticals, 2022, doi:10.3390/ph15101187_

Round 1

Reviewer 1 Report

The current manuscript studied the effect of capsaicin on several voltage-gated ion channel currents in rabbit ventricular cardiomyocytes by patch-clamp. They found that capsaicin inhibited rapidly IKr and slowly IKs K+ currents and Ito K+ current with lower IC50 values, and suppressed voltage-gated Na+ and Ca2+ currents and inward rectifier IK current at higher concentrations. They suggested that the effect of capsaicin on cardiac electrophysiology is due to its inhibition for these voltage-gated ion channels. I have several concerns to get such conclusion.

1.     It may not sufficient to identify different voltage-gated ion channel by stimulate protocols. For potassium channel, known channel blocker XE991 may be needed to identify potassium channel. Alternatively, overexpression of these voltage-gated channel in tool cells may further to confirm the effect from capsaicin.

2.      Capsaicin is known activator for TRPV1. In the manuscript, authors used capsazepine, the antagonist of TRPV1 to exclude the role of TRPV1. However, as the author noticed, “The inhibitory effects of capsaicin on these currents developed gradually reaching steady-state levels within 3 to 6 min”, the inhibition of capsaicin may indirectly act to the channels. Therefore, it is important to prove that the effect of capsaicin is unrelated to TRPV1 activation.  Authors may consider test if temperature or proton will alter the inhibition of capsaicin for voltage-gated channels.

3. All the maximum inhibition of capsaicin for different voltage gated ion channels are 80-90%. That is unusual. How to explain they have similar maximum inhibition? 

Author Response

September 5, 2022

Re: Manuscript ID: pharmaceuticals-1851758

Dear Editor,

Thank you for your review of our manuscript number 821758 entitled “Capsaicin directly inhibits multiple voltage-gated ion channels in rabbit ventricular cardiomyocytes” We appreciate the opportunity to address the concerns of the reviewers and have made the appropriate changes according to suggestions of reviewers as described below.

Reply to the comments of reviewer 1:

  1. It may not sufficient to identify different voltage-gated ion channel by stimulate protocols. For potassium channel, known channel blocker XE991 may be needed to identify potassium channel. Alternatively, overexpression of these voltage-gated channel in tool cells may further to confirm the effect from capsaicin.

REPLY: We thank reviewer for his/her valuable comments. Pharmacological and functional characterization of IKr and IKs were described in the methods section. We have used Dofetilide (3 µM), an established specific blocker of IKr and HMR 1556 (3 µM), specific blocker of IKs. In initial studies, Dofetilide and HMR 1556 effectively blocked IKr and IKs, respectively, indicating potassium currents under investigation are indeed IKr and IKs. This information was provided in the methods section (page 14, lines 391-395).

  1. Capsaicin is known activator for TRPV1. In the manuscript, authors used capsazepine, the antagonist of TRPV1 to exclude the role of TRPV1. However, as the author noticed, “The inhibitory effects of capsaicin on these currents developed gradually reaching steady-state levels within 3 to 6 min”, the inhibition of capsaicin may indirectly act to the channels. Therefore, it is important to prove that the effect of capsaicin is unrelated to TRPV1 activation. Authors may consider test if temperature or proton will alter the inhibition of capsaicin for voltage-gated channels.

REPLY: We thank reviewer for pointing out this important issue. As the reviewer pointed out, we have used an established TRPV1 receptor antagonist to rule out the involvement of TRPV1 channels. More importantly application of capsaicin alone did not induce typical capsaicin-induced inward currents. Furthermore, several studies have reported that rodent cardiomyocytes do not expresses TRPV1 channels. This information was stated and related references were cited in discussion section (page 10, lines 263-269)

  1. All the maximum inhibition of capsaicin for different voltage gated ion channels are 80-90%. That is unusual. How to explain they have similar maximum inhibition?

REPLY: we than reviewer for his/her comments. Full inhibition of different voltage-gated ion channels by capsaicin is not an unusual feature. High efficacy (80-90% inhibition) of capsaicin on various ion channels has been in several earlier studies. In fact, low efficacy at high concentrations of these lipophilic drugs including capsaicin, endogenous cannabinoids, steroids, general anesthetics and other lipophilic compounds indicated in the discussion section.

Castle, N.A. Differential inhibition of potassium currents in rat ventricular myocytes by capsaicin. Cardiovasc Res 1992, 26, 1137-1144, doi:10.1093/cvr/26.11.1137.

Wu, S.N.; Chen, I.J.; Lo, Y.C.; Yu, H.S. The characteristics in the inhibitory effects of capsaicin on voltage-dependent K(+) currents in rat atrial myocytes. Environ Toxicol Pharmacol 1996, 2, 39-47, doi:10.1016/1382-6689(96)00028-2.

Xing, J.; Ma, J.; Zhang, P.; Fan, X. Block effect of capsaicin on hERG potassium currents is enhanced by S6 mutation at Y652. Eur J Pharmacol 2010, 630, 1-9, doi:10.1016/j.ejphar.2009.11.009.

Alzaabi, A.H.; Howarth, L.; El Nebrisi, E.; Syed, N.; Susan Yang, K.H.; Howarth, F.C.; Oz, M. Capsaicin inhibits the function of α(7)-nicotinic acetylcholine receptors expressed in Xenopus oocytes and rat hippocampal neurons. Eur J Pharmacol 2019, 857, 172411, doi:10.1016/j.ejphar.2019.172411.

Nebrisi, E.E.; Prytkova, T.; Lorke, D.E.; Howarth, L.; Alzaabi, A.H.; Yang, K.S.; Howarth, F.C.; Oz, M. Capsaicin Is a Negative Allosteric Modulator of the 5-HT(3) Receptor. Front Pharmacol 2020, 11, 1274, doi:10.3389/fphar.2020.01274.

Reviewer 2 Report

The authors show that capsaicin inhibits multiple voltage-gated ion channels. Although this paper is very interesting and carefully conducted, it would be accepted for publication in Pharmaceutics if the following point is addressed.

The t-test can be used between two groups, but not for three or more groups. Please perform the statistics again.

Author Response

September 5, 2022

Re: Manuscript ID: pharmaceuticals-1851758

Dear Editor,

Thank you for your review of our manuscript number 821758 entitled “Capsaicin directly inhibits multiple voltage-gated ion channels in rabbit ventricular cardiomyocytes” We appreciate the opportunity to address the concerns of the reviewers and have made the appropriate changes according to suggestions of reviewers as described below.

Reply to the comments of reviewer 2:

The t-test can be used between two groups, but not for three or more groups. Please perform the statistics again.

REPLY: We thank reviewer for pointing out this important issue. We have reevaluated our data using ANOVA method and provided this information in figure legends of figures 1-6.

Reviewer 3 Report

author should improve introduction and must be oriented with effect of capsaicin on heart and  mechanism of action of capsaicin on heart rate and blood pressure. bridging of literature with your propose objective needed. Conclusion must be according to objectives made in introduction.

In introduction line no. 57.

1. Capsaicin acts through TRPV1 channels but what compelled you to study Na and Ca++ channels. You need to give some more literature on different mechanism of Capsaicin on heart and link it with your objective by bridging area of gap.

2. Literature is missing about effects of capsaicin on the heart rate, blood pressure and possible involvement of receptors.

3. literature related to capsaicin on heart and latest speculation about involvement of pathways must mention here to bridge your objectives of the study.

Material method:

2. 100% DMSO is toxic and carcinogenic. Did you check solubility of capsaicin in other solvents or titration of DMSS with other solvent?

3. write all range of doses selected for capsaicin or at least min and max

Result:

4. would it not be a great idea to use 3 doses of capsaicin?

5. what was negative and positive control in this experiment?

6. Statistical significance in Fig. 4-C should be labelled

7. Statistical significance in Fig. 5-C should be labelled

Discussion:

8. At line number 231, does it correlate with beta receptors inhibition or CB1 and CB2 receptors on heart in literature?

9. at line 244, can we speculate that Capsaicin produces its action via ligand gated ion channels of receptor family? can we extend discussion to this direction?

10. at line 271, it might be interaction with CB1 and CB2 receptors as capsaicin produces its anti-inflammatory effects via CB2 receptors and CB1 and CB2 receptor expressed in heart. I suggest give explanation over there.

11. At line number 322, Concentration in the brain does have any connection with ion channels of cardiomyocyte?

12. give cascade of pathway that what will happen to cardiovascular parameters when capsaicin will actson these slow and rapid channels of K and Ca++

Conclusion:

12. Conclusion should be concrete, specific and in line with the objectives of your study.

Author Response

September 5, 2022

Re: Manuscript ID: pharmaceuticals-1851758

Dear Editor,

Thank you for your review of our manuscript number 821758 entitled “Capsaicin directly inhibits multiple voltage-gated ion channels in rabbit ventricular cardiomyocytes” We appreciate the opportunity to address the concerns of the reviewers and have made the appropriate changes according to suggestions of reviewers as described below.

Reply to the comments of reviewer 3:

author should improve introduction and must be oriented with effect of capsaicin on heart and  mechanism of action of capsaicin on heart rate and blood pressure. bridging of literature with your propose objective needed. Conclusion must be according to objectives made in introduction.

REPLY: We thank reviewer for his/her comments. In line with the suggestions of the reviewer. We have provided more information and cited relevant references on the capsaicin and blood pressure in the introduction section (pages 1 and 2, lines 44-47). 

In introduction line no. 57.

  1. Capsaicin acts through TRPV1 channels but what compelled you to study Na and Ca++ channels. You need to give some more literature on different mechanism of Capsaicin on heart and link it with your objective by bridging area of gap.

REPLY: We thank reviewer for his/her comments. This is an electrophysiological study intended to investigate the effects of capsaicin on major voltage-gated ion channels in cardiac myocytes. Voltage-gated Na and Ca2+ channels are prominent inward currents activated by voltage changes in cardiomyocytes and therefore these channels, in addition to K+ channels were the target of our investigation. As it was mentioned in the discussion, the effects of capsaicin on some of these channels were already investigated in different cell types. However, this is the first study to comparatively investigate these channels in cardiomyocytes.

  1. Literature is missing about effects of capsaicin on the heart rate, blood pressure and possible involvement of receptors.

REPLY: We have provided sufficient literature on capsaicin effect on contractility and cardiac action potential characteristics in the introduction. In agreement with the reviewers suggestion, we have incorporated more information regarding the effects of capsaicin on blood pressure and provided major references in the introduction section (Pages 1 and 2, lines 44-47).   

  1. literature related to capsaicin on heart and latest speculation about involvement of pathways must mention here to bridge your objectives of the study.

REPLY: We thank reviewer for his/her comments. Our study focuses on the effects of capsaicin on the functions of voltage-gated ion channels in cardiomyocytes. We have covered published literature covering the effects of capsaicin on cardiomyocytes and to some extent on the heart. In addition, in the introduction we have provided more literature reviewing the effects of capsaicin on the vascular structures and blood pressure. We are not aware of “latest speculations about the involvement of pathways”. We would be happy to include further information if a specific suggestion was provided on this subject.    

Material method:

  1. 100% DMSO is toxic and carcinogenic. Did you check solubility of capsaicin in other solvents or titration of DMSS with other solvent?

REPLY: We agree with the reviewer in that 100% DMSO is cytotoxic and carcinogenic. In DMSO stock solutions, capsaicin (10-30 mM) was completely dissolved without any precipitation. As we mentioned in the methods section of our manuscript (page 14, lines 405-406), DMSO was diluted in average 1000-fold, and the highest final concentration of DMSO in the extracellular solutions was 0.03% (v/v). At this diluted concentration, DMSO did not affect the amplitudes of the membrane currents investigated in this study. This information was stated in the methods section (page 14; lines 405-406).

  1. write all range of doses selected for capsaicin or at least min and max

REPLY: We have presented 6 dose-response curves for the 6 voltage-gated ion channels investigated in this study. Concentration ranges of capsaicin (both maximum and minimum concentrations of capsaicin) are clearly indicated in each dose-response (concentration-inhibition) curve (Figure 1E, 2E, 3E, 4E, 5E, and 6E). 

Result:

  1. would it not be a great idea to use 3 doses of capsaicin?

REPLY: In average we have used  5 to 6 different concentrations to construct a dose-response (concentration-inhibition) curve with minimum and maximum effects are clearly shown. In order to construct a reliable concentration-inhibition curve to calculate IC50 values, these number of concentrations are needed.

  1. what was negative and positive control in this experiment?

REPLY: By definition, Positive Control is an experimental control that gives a positive result at the end of the experiment. Negative Control is an experimental control that does not give a response to the test. In electrophysiological studies, the classical  experimental design necessitates before (which constitutes control values) and after drug application during patch-clamp studies. However, in our preliminary studies, we have verified the identity of each channel by pharmacological and functional experiments and we have provided this information in the methods section (page 14, lines 3943-398).

  1. Statistical significance in Fig. 4-C should be labelled

REPLY: Figure 4C presents only raw data derived from current recording at given potential. Statistical treatment of the same data were presented in Figure 4D and this information was provided in the figure legend. In the figure 4D, the extent of inhibition at different potentials is 45 to 55 % and obviously significant inhibition.  

  1. Statistical significance in Fig. 5-C should be labelled

REPLY: Figure 5C presents only raw data derived from current recording at given potential. Statistical treatment of the same data were presented in Figure 5D and this information was provided in the figure legend. In the figure 5D, the extent of inhibition at different potentials is 45 to 55 % and obviously significant inhibition. 

Discussion:

  1. At line number 231, does it correlate with beta receptors inhibition or CB1 and CB2 receptors on heart in literature?

REPLY: To the best of our knowledge, there is no literature indicating that capsaicin binds to beta adrenergic receptors. For cannabinoid receptors, capsaicin has been shown to bind CB1 and CB2 receptors with low affinity (affinity higher than 10 µM). Since we have used capsaicin concentrations higher than 10 µM, we tested SR141716A (2 µM), a specific CB1 antagonist, and SR 144528 (2 µM), a specific CB2 antagonist on capsaicin inhibition of INa. The extent of capsaicin inhibition of INa was not altered in the presence of these antagonists. This information was provided in the discussion section (page 10, lines 270-275). 

Melck, D.; Bisogno, T.; De Petrocellis, L.; Chuang, H.; Julius, D.; Bifulco, M.; Di Marzo, V. Unsaturated long-chain N-acyl-vanillyl-amides (N-AVAMs): vanilloid receptor ligands that inhibit anandamide-facilitated transport and bind to CB1 cannabinoid receptors. Biochem Biophys Res Commun 1999, 262, 275-284, doi:10.1006/bbrc.1999.1105.

  1. at line 244, can we speculate that Capsaicin produces its action via ligand gated ion channels of receptor family? can we extend discussion to this direction?

REPLY: By definition, ligand-gated ion channels are opened by specific ligands but not voltage changes. Therefore, it is unlikely that ligand-gated ion channels are involved in voltage protocols used in our study. However, 5-HT3 receptors and nicotinic acetylcholine receptors expressed in Xenopus oocytes are inhibited by capsaicin (Alzaabi et al., 2019; Nebrisi et al., 2020).

Alzaabi, A.H.; Howarth, L.; El Nebrisi, E.; Syed, N.; Susan Yang, K.H.; Howarth, F.C.; Oz, M. Capsaicin inhibits the function of α(7)-nicotinic acetylcholine receptors expressed in Xenopus oocytes and rat hippocampal neurons. Eur J Pharmacol 2019, 857, 172411, doi:10.1016/j.ejphar.2019.172411.

Nebrisi, E.E.; Prytkova, T.; Lorke, D.E.; Howarth, L.; Alzaabi, A.H.; Yang, K.S.; Howarth, F.C.; Oz, M. Capsaicin Is a Negative Allosteric Modulator of the 5-HT(3) Receptor. Front Pharmacol 2020, 11, 1274, doi:10.3389/fphar.2020.01274.

  1. at line 271, it might be interaction with CB1 and CB2 receptors as capsaicin produces its anti-inflammatory effects via CB2 receptors and CB1 and CB2 receptor expressed in heart. I suggest give explanation over there.

REPLY: In the literature, there is one study (Melck et al., 1999) indicating that capsaicin can bind CB1 and CB2 receptors with low affinity (affinity higher than 10 µM). Since we have used capsaicin concentrations higher than 10 µM, we tested SR141716A (2 µM), a specific CB1 antagonist, and SR 144528 (2 µM), a specific CB2 antagonist on capsaicin inhibition of INa. The extent of capsaicin inhibition of INa was not altered in the presence of these antagonists. This information was provided in the discussion section (page 10, lines 270-275).  

Melck, D.; Bisogno, T.; De Petrocellis, L.; Chuang, H.; Julius, D.; Bifulco, M.; Di Marzo, V. Unsaturated long-chain N-acyl-vanillyl-amides (N-AVAMs): vanilloid receptor ligands that inhibit anandamide-facilitated transport and bind to CB1 cannabinoid receptors. Biochem Biophys Res Commun 1999, 262, 275-284, doi:10.1006/bbrc.1999.1105.

  1. At line number 322, Concentration in the brain does have any connection with ion channels of cardiomyocyte?

REPLY: We agree with the reviewer in that brain concentrations of capsaicin should be different than heart. However, there is no literature studying the capsaicin concentrations in cardiac tissue and, we believe that in the absence of any literature, it would be useful to provide available literature on the tissue concentrations of capsaicin.   

  1. give cascade of pathway that what will happen to cardiovascular parameters when capsaicin will actson these slow and rapid channels of K and Ca++

REPLY: We are not sure which cardiovascular parameters were referred to by the reviewer. However, possible cardiac contractility and cardiac action potential changes as a result of capsaicin inhibition of these ion channels were discussed in the discussion section and computer modeling of these parameters were presented in the figure 7 and figure 8. Furthermore, additional information and related references regarding the capsaicin-induced blood pressure changes were provided in the introduction section of the revised manuscript (page 1, line 44-47).

Conclusion:

  1. Conclusion should be concrete, specific and in line with the objectives of your study.

REPLY: We thank the reviewer for his/her valuable comments. We have made some necessary changes in the introduction section on blood pressure (page 1, lines 44-47) and on the issue of cannabinoid receptors in the discussion section (page 10, lines 270-275) of the manuscript.    

Round 2

Reviewer 1 Report

The authors addressed all my concerns.

Author Response

We thank reviewer 1 for his/her valuable comments